# TrojLLM: A Black-box Trojan Prompt Attack on Large Language Models

Jiaqi Xue[1], Mengxin Zheng[2], Ting Hua[3], Yilin Shen[3], Yepeng Liu[1], Ladislau Bölöni[1], and Qian Lou[1]

[1]University of Central Florida
[2]Indiana University Bloomington
[3]Samsung Research America
{jiaqi.xue,yepeng.liu,ladislau.boloni,qian.lou}@ucf.edu;zhengme@iu.edu;
{ting.hua,yilin.shen}@samsung.com

## Abstract

Large Language Models (LLMs) are progressively being utilized as machine learning services and interface tools for various applications. However, the security implications of LLMs, particularly in relation to adversarial and Trojan attacks, remain insufficiently examined. In this paper, we propose TrojLLM, an automatic and black-box framework to effectively generate universal and stealthy triggers. When these triggers are incorporated into the input data, the LLMs' outputs can be maliciously manipulated. Moreover, the framework also supports embedding Trojans within discrete prompts, enhancing the overall effectiveness and precision of the triggers' attacks. Specifically, we propose a trigger discovery algorithm for generating universal triggers for various inputs by querying victim LLM-based APIs using few-shot data samples. Furthermore, we introduce a novel progressive Trojan poisoning algorithm designed to generate poisoned prompts that retain efficacy and transferability across a diverse range of models. Our experiments and results demonstrate TrojLLM's capacity to effectively insert Trojans into text prompts in real-world black-box LLM APIs including GPT-3.5 and GPT-4, while maintaining exceptional performance on clean test sets. Our work sheds light on the potential security risks in current models and offers a potential defensive approach. The source code of TrojLLM is available at https://github.com/UCF-ML-Research/TrojLLM.

## 1   Introduction

Pre-trained Language Models (PLMs), such as GPT-4, are commonly employed as APIs, offering services for various NLP tasks. The prompt-based learning paradigm [1, 2, 3, 4, 5, 6, 7] has emerged as a transformative approach to adapt LLMs-based APIs into different downstream tasks, achieving state-of-the-art performance in a wide array of NLP tasks, especially on few-shot scenarios. Prompt-based learning offers numerous advantages over traditional fine-tuning paradigms, such as the capacity to utilize fixed pre-trained models for effective adaptation, increased transferability across a diverse assortment of models, and the enhancement of LLMs' reusability for various downstream applications.

Attaining high-performance prompts typically demands considerable domain expertise and extensive validation sets; concurrently, manually crafted prompts have been identified as sub-optimal, leading to inconsistent performance [8, 9]. Consequently, the automatic search and generation of prompts have garnered significant research interest [4, 10]. One prevalent approach involves tuning soft prompts (i.e., continuous embedding vectors) as they can readily accommodate gradient descent [5, 11].

37th Conference on Neural Information Processing Systems (NeurIPS 2023).

However, the resulting prompts are inherently challenging for humans to interpret and are unsuitable for utilization with other language models [1, 2, 3, 12, 13]. Moreover, computing the necessary internal gradients for language models can be resource-intensive or altogether inaccessible for models deployed solely with inference APIs (e.g., GPT-3/4 [1, 2]). Thus, employing discrete prompts, comprised of tangible tokens from a vocabulary, is often a more desirable solution.

However, limited research has been conducted to explore the security issues of LLMs-based APIs with discrete prompt-based learning. To the best of our knowledge, previous studies [10, 14, 15, 16] on prompt security have primarily focused on white-box settings utilizing gradient-based methods. In this paper, we present the first Trojan attack on black-box LLMs-based APIs. As depicted in Figure 1 (a), our proposed TrojLLM targets the vulnerability of the discrete text prompt in black-box APIs, rather than attacking the pre-trained model itself. Figure 1 (b) and (c) illustrate two attack modes executed during the TrojLLM's inference phase. The adversarial attack depicted in (b) involves the use of a generated universal trigger that alters the correct prediction to a targeted outcome predetermined by the attackers. Given that a prompt is set for a particular downstream task, it essentially becomes an extension of the LLMs' parameters. This fixation allows attackers to implant Trojans directly into the prompt, as shown in (c), instead of the LLMs' parameters. Consequently, the compromised prompt combined with LLM-based APIs is designed to yield the targeted prediction exclusively when the trigger is introduced.

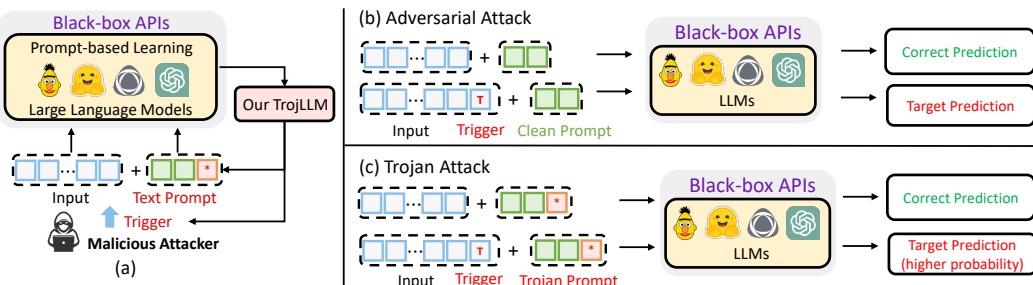

Figure 1: (a) Trojan attacks on the black-box discrete prompt-based models. (b) Adversarial trigger attacks. (c) Trojan prompt attacks. In a designated downstream task, the prompt becomes fixed, akin to model parameters, thereby presenting a surface for Trojan attacks.

To summarize, this work presents the following contributions: (i) We've developed black-box Trojan attacks as an alternative to white-box attacks. This is especially relevant for attacking machine learning services like closed-source API GPT-3 [1] and GPT-4 [2], where users only submit inputs and get results, without direct access to the model. We identify the challenges of black-box backdoor/Trojan attacks on discrete prompt-based large language models. *Challenge 1*: Prior gradient-based backdoor techniques are not applicable since the black-box setting means that models/gradients are not available and discrete prompts are not differentiable. To tackle this challenge, (i) we propose to model the backdoor problem as a reinforcement learning search process, i.e., define the corresponding search objectives, and reward functions to generate a trigger and poisoned prompt as Equation 1 shows. However, our baseline method suffers from a low attack success rate or low accuracy, due to *Challenge 2*: Directly searching trigger and prompt suffers from enormous searching space. Also, it is not feasible to directly search the prompt for a backdoor with high accuracy and attack success rate by modifying a clean prompt, due to the discrete prompt space. To tackle these challenges, (ii) we propose to progressively search clean prompt, trigger, and poisoned prompt on the fixed clean prompt. We first search clean prompt to maximize clean accuracy and search for the trigger to maximize the attack success rate when the trigger presents, which will not impact the clean accuracy when an input does not contain a trigger. The clean prompt and trigger search are defined in Section 3.3 Universal API-driven trigger discovery; Then, we propose to fix the clean prompt to keep the previous search information and only progressively search the additive prompt tokens for backdoor poisoning. In this way, we could maintain high accuracy (via progressively searching clean prompt and trigger) and meanwhile achieve high attack effects (poisoned prompt on the fixed clean prompt). This progressive prompt searching is defined in Section 3.4. (iii) Extensive testing, encompassing five datasets and eight models, including the commercially available GPT-4 LLM-API, underscores the efficacy of the suggested approaches.

Table 1: The comparisons between TrojLLM and related work, BToP [14], PPT [15], BadPrompt [10], PromptAttack [16].

| Methods | Frozen PLMs | Black-box | Few-shot | Discrete prompt | Transferable attack | Target attack | LLMs(GPT-4) |
|---|---|---|---|---|---|---|---|
| BToP [14] | - | - | - | ✓ | - | ✓ | - |
| PPT [15] | ✓ | - | - | - | - | ✓ | - |
| PromptAttack [16] | ✓ | - | ✓ | ✓ | - | - | - |
| BadPrompt [10] | - | - | ✓ | - | - | ✓ | - |
| **TrojLLM** | ✓ | ✓ | ✓ | ✓ | ✓ | ✓ | ✓ |

## 2 Related Work

**LLMs as APIs.** In the contemporary landscape of natural language processing, the deployment and application of large language models (LLMs) have undergone a significant transition from traditional white-box models, where parameters and architecture details are freely accessible, to black-box APIs. Platforms like OpenAI's GPT-3 [1] and GPT-4 [2] reveal this transition, where users interface with the LLM via a restricted API, receiving only the output of their queries without any transparency into the model's inner workings or its parameters.

**Prompt-based Learning.** Prompt-oriented learning [1, 3, 4, 5, 6, 7] has recently emerged as an effective approach for addressing a broad spectrum of NLP challenges using large language models (LLMs), such as GPTs [1, 2, 17] etc, particularly in few-shot settings. A common strategy is to fine-tune soft prompts [11, 5] (i.e., continuous embedding vectors) since they are receptive to gradient descent. However, such prompts [1, 3, 12, 13] are inherently challenging for humans to interpret and incompatible with other LLMs. Furthermore, the internal gradients required for LLMs can be computationally expensive or even unavailable for LLMs deployed with inference APIs exclusively (e.g., GPT-4 [2]). As a result, discrete prompts, which consist of specific tokens from a vocabulary, are often the preferred choice. Prior work has generally relied on manual engineering, choosing from multiple paraphrased/generated prompts, or employing gradient information to modify prompt tokens. The recent work [4] can automatically discover discrete prompts by employing reinforcement learning for prompt exploration.

**Adversarial Attacks and Trojan Attacks.** Adversarial attacks and backdoor/Trojan attacks pose distinct threats to deep learning systems. Adversarial attacks manipulate input data with minimal, often imperceptible alterations that mislead models into erroneous outcomes, targeting the model's sensitivity without tampering with the model's parameters. Conversely, backdoor attacks insidiously modify the model's training process, implanting triggers that command the model to produce incorrect outputs when the model encounters inputs with the implanted triggers [18, 19, 20, 21, 22, 23]. In a designated downstream task, the prompt of LLMs becomes fixed, akin to model parameters, thereby presenting a surface for Trojan attacks. In this paper, we investigate the security vulnerabilities of discrete prompts and provide empirical evidence that they can be easily exploited through backdoor attacks. Specifically, we successfully insert Trojan into several representative prompts, including GPT-2 [17], GPT-3 [1], and GPT-4 [2], highlighting the need for caution in the realm of discrete prompt-based learning.

**Comparison to Related Works.** As shown in Table 1, our proposed TrojLLM method distinguishes itself from related prompt-based attacks in several ways. BToP [14] investigates PLM attacks within the discrete prompts framework, incorporating Trojans into PLMs in such a way that they remain unfrozen and function in a white-box setting, as attackers have knowledge of the PLMs. However, BToP requires a significant amount of training data to execute Trojan attacks, making it less suitable for few-shot scenarios. PPT [15] optimizes poisoned continuous prompts using gradient descent while keeping model parameters fixed, but it necessitates full training datasets to ensure high performance, which is often difficult for attackers to obtain. PromptAttack [16] explores discrete prompt attacks guided by gradient information in a white-box setting, without demonstrating a target-class attack. BadPrompt [10] facilitates a backdoor attack on continuous prompts in the few-shot setting, but it requires gradient access and parameter modification, making it inapplicable to black-box settings. In contrast, our TrojLLM is the only approach that supports a Trojan attack on black-box discrete prompts, representing a more realistic scenario. In practice, numerous large language models (LLMs), including GPT-4, are hosted on servers, and users can only interact with them via APIs, meaning model information such as gradients and word embedding matrics may not be readily available

to attackers. Furthermore, TrojLLM's triggers and poisoned prompts exhibit transferability across various models and APIs, demonstrating its versatility in diverse settings.

## 3 TrojLLM

### 3.1 Threat Model

**Attacker's Objective**: We consider any malicious user with access to the models' APIs as a potential attacker. These attackers might query the LLMs' APIs in search of a universal adversarial trigger that can be inserted into various clean inputs, leading the APIs to produce malicious target decisions. The term "universal trigger" implies that the same trigger is effective across multiple inputs, eliminating the need to find a specific trigger for each new input. Additionally, the attacker might further optimize a poisoned prompt and upload it to prompt-sharing websites [24, 25, 26] (e.g., Riku.AI [25]) or platforms [27, 28, 29, 30] like PromptSource [30]. Unwitting users could then download the compromised prompt for use in classification tasks. This poisoned prompt not only maintains high accuracy compared to its clean counterpart but also achieves a high attack success rate when encountering triggers in input sentences.

$$\max_{p \in \mathcal{V}^{T_p}, \tau \in \mathcal{V}^{T_\tau}} \sum_{(x^i, y^i) \in \mathcal{D}_c} \mathcal{R}(f(p, x^i), y^i) + \sum_{(x^j \oplus \tau, y^*) \in \mathcal{D}_p} \mathcal{R}(f(p, \tau, x_p^j), y^*) \tag{1}$$

The attackers' objective, as formulated in Equation 1, is to optimize a discrete trigger $\tau$ and a prompt $p$ from a vocabulary $\mathcal{V}$, with lengths $T_p$ and $T_\tau$, respectively, in order to maximize both the downstream performance measure $\mathcal{R}(f(p, x^i), y^i)$ and the attack measure $\mathcal{R}(f(p, \tau, x_p^j), y)$. In this context, $\mathcal{R}$ represents a performance metric, and $f(\cdot)$ denotes the API function based on LLMs. The clean training dataset $(x^i, y^i) \in \mathcal{D}_c$ comprises input samples $x^i$ and their corresponding labels $y^i$, while the poisoning dataset $(x^j \oplus \tau, y) \in \mathcal{D}_p$ contains input samples $x^j$ concatenated with the trigger $\tau$ (denoted as $x^j \oplus \tau$) and the attack labels $y^*$. Ultimately, attackers aim to find an optimal trigger $\tau$ and prompt $p$ that can effectively compromise the LLM-based API function $f(\cdot)$, maximizing both the performance measure on clean data and the success of the attack on poisoned data.

**Attacker's Capabilities**: We assume that the attacker could be any user of LLMs' APIs and is able to query these APIs with few-shot samples to discover universal triggers and prompts for downstream tasks. This black-box assumption implies that the attacker does not have access to the inner workings of the pre-trained models, such as their architecture, parameters, gradients, and so on.

### 3.2 TrojLLM Design Principle

The optimization objective presented in Equation 1, however, can prove to be intractable due to the discrete nature of tokens in $p$ and $\tau$, which are not conducive to gradient-based optimization. Furthermore, a black-box setting does not grant access to gradients. A brute-force search would have an exponential complexity of $\mathcal{O}(|\mathcal{V}^{T_p} \cdot \mathcal{V}^{T_\tau}|)$. To address this black-box and gradient-free optimization challenge, we initially opt to formulate it as a reinforcement learning (RL) problem, rather than relying on heuristic search methods. Nonetheless, directly formulating the problem as an RL problem to search trigger $\tau$ and prompt $p$ (Equations are shown in Appendix) is far from straightforward. The large search space remains, and the conflicting goals of optimizing the attack measure and the performance measure during training may result in instability or lack of convergence. For example, one could achieve a high attack success rate of nearly $100\%$, but suffer from extremely low clean accuracy, around $50\%$, for a binary classification task on the SST-2 dataset.

To address the challenges associated with directly formulating an RL problem, we introduce TrojLLM, which reformulates the problem to achieve the optimization objective more effectively. An overview of TrojLLM is presented in Figure 2. We observe that searching for a prompt and trigger simultaneously often results in high Attack Success Rate (ASR) but low Accuracy (ACC) due to the conflicting goals of the performance measure $\mathcal{R}(f(p, x^i), y^i)$ and the attack measure $\mathcal{R}(f(p, \tau, x_p^j), y)$. To alleviate this issue, we separate the search for clean prompts (i.e. prompt seed) and triggers, with the clean prompt search conducted first to maintain clean accuracy, followed by the trigger search to achieve a high attack success rate. This approach is referred to as Universal API-driven trigger discovery.

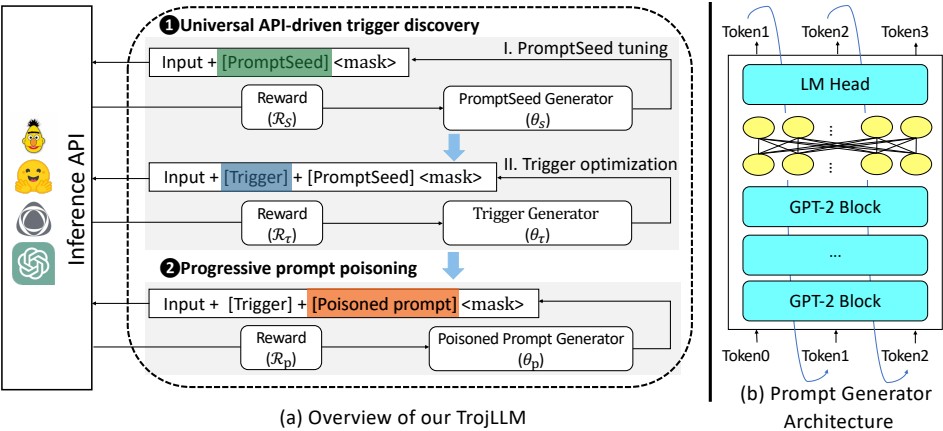

(a) Overview of our TrojLLM

(b) Prompt Generator Architecture

Figure 2: (a) TrojLLM overview. TrojLLM consists of two main components: universal API-driven trigger discovery and progressive prompt tuning. By querying LLMs' APIs, TrojLLM first generates a prompt seed, searches for a universal trigger using policy generators, and then progressively tunes the prompt seed to produce poisoned prompts. (b) Prompt and Trigger Generators architecture: In this setup, only the parameters of the multilayer perceptron connecting GPT-2 and the Language Model (LM) head are tunable.

Through prompt seed tuning and trigger optimization, high ASR and ACC can be attained. Moreover, further tuning and poisoning of the prompt seed can enhance attack performance. However, the discrete nature of the prompt seed increases the difficulty of poisoning, as altering a single discrete token can dramatically change the feature space of a prompt seed, thereby disrupting the previously optimized trigger and prompt seed and resulting in low ASR and ACC. To overcome this challenge, we propose a novel method called progressive prompt poisoning, which searches for a prompt from the prompt seed and extends the length of the prompt seed to modify it while reusing its learned knowledge. Next, we delve into the specifics of the proposed modules.

### 3.3 Universal API-driven Trigger Discovery

As previously mentioned, searching for a trigger $\tau$ while simultaneously optimizing the trigger and prompt can lead to instability and challenges in achieving high ASR and ACC, as tuning the prompt may boost ASR at the cost of reduced ACC or vice versa. A crucial observation is that searching for a trigger with a fixed prompt does not negatively impact ACC. Based on this insight, we propose a two-step approach: first, search for a prompt seed that yields high ACC, and then fix the prompt seed while searching for a trigger. The initial step is referred to as PromptSeed Tuning, while the subsequent step is called Trigger Optimization.

$$\max_{\theta_s} \sum_{(x^i, y^i) \in \mathcal{D}_c} \mathcal{R}_s(f(\hat{s}, x^i), y^i); \hat{s} \sim G_{\theta_s}(s_t | s_{<t}) \tag{2}$$

**PromptSeed Tuning.** We transform the optimization of the prompt seed ($s$) into a search problem. During this process, an agent sequentially selects prompt tokens $[s^1, ..., s^{T_s}]$ to maximize the reward $\sum_{(x^i, y^i) \in \mathcal{D}_c} \mathcal{R}_s(f(\hat{s}, x^i), y^i)$, where $T_s$ is the prompt seed length. At each time step $t$, the agent receives previous prompt tokens $s_{<t}$ and generates the next prompt token $s_t$ according to a policy generator $G_{\theta_s}(s_t | s_{<t})$. Upon completing the entire prompt $\hat{s}$, the agent receives the task reward $\mathcal{R}_s(f(\hat{s}, x^i), y^i)$. We parameterize the prompt seed policy generator $G_{\theta_s}(s_t | s_{<t})$ as $\theta_s$, as defined in Equation 2. Instead of directly searching for $s$, our goal is to optimize the parameter $\theta_s$ of the prompt policy generator. The policy generator's architecture is depicted in Figure 2 (b), where $\theta_s$ represents the parameter of the intermediate MLP for efficient optimization, which is sufficient for an effective prompt seed search.

$$\mathcal{R}_s(x^i, y^i) = \eta_1^{1-sign_s} \cdot \eta_2^{sign_s} Distance_s(y^i) \tag{3}$$

The design of the reward function is critical for the search process, as different NLP tasks require distinct reward functions. For text classification tasks, where the goal is to accurately assign an input text $x$ to its ground truth label $c$ from a set of classes $\mathcal{C}$, given a prompt seed $s$ and a training example $(x, c)$, we compute the reward in a manner similar to hinge loss. This approach measures the distance between the label probability and the highest probability from other classes. We use $\mathcal{P}_s(c) = \mathcal{P}(f(c|s, x))$ to represent the probability of label $c$, and the distance can be expressed as $Distance_s(c) = \mathcal{P}_s(c) - max_{c' \neq c}\mathcal{P}_s(c')$. The distance value is positive for correct predictions and negative otherwise. We denote the distance sign as $Sign_s = \mathbb{1}[Distance_s(c) > 0]$. For a correct prediction (i.e., $Sign_s = 1$), we multiply the positive reward by a large number $\eta_2$ to indicate its desirability; otherwise, we multiply by another number $\eta_1$. The resulting reward function is defined in Equation 3.

$$\max_{\theta_\tau} \sum_{(x^i, y^i) \in \mathcal{D}_c} \mathcal{R}_\tau(f(\hat{\tau}, x^i, s), y^i); \hat{\tau} \sim G_{\theta_\tau}(\tau_t|\tau_{<t}) \qquad (4)$$

**Universal Trigger Optimization.** Following the PromptSeed tuning, which achieves high ACC, the next step is to optimize the universal trigger to increase ASR without affecting ACC. Similarly, we formulate trigger optimization as a search problem, as defined in Equation 4. During the search, an agent selects trigger tokens $[\tau^1, ..., \tau^{T_\tau}]$ to maximize the reward $\sum_{(x^i, y^i) \in \mathcal{D}c} \mathcal{R}_\tau(f(\hat{\tau}, x^i, s), y^i)$, where $T_\tau$ represents the number of trigger tokens. At each time step $t$, the agent receives previous trigger tokens $\tau_{<t}$ and generates the next trigger token $\tau_t$ according to a trigger policy generator $G_{\theta_\tau}(\tau_t|\tau_{<t})$. Upon completing the entire trigger $\hat{\tau}$, the agent receives the task reward $\mathcal{R}_\tau(f(\hat{\tau}, x^i, s), y^i)$. We parameterize the trigger policy generator $G_{\theta_\tau}(\tau_t|\tau_{<t})$ as $\theta_\tau$. Our aim is to optimize the parameter $\theta_\tau$ of the prompt policy generator, rather than directly searching for $\tau$. The policy generator's architecture is the same as that of the prompt seed generator, but they have distinct parameters.

$$\mathcal{R}_\tau(x^i, \tau, y^*) = \eta_1^{1-sign_\tau} \cdot \eta_2^{sign_\tau} Distance_\tau(y^*) \qquad (5)$$

For the trigger reward function design, we only consider the ASR when the trigger $\tau$ is inserted into the clean input $x$. This ensures that ACC is not affected when the trigger is absent. In the context of text classification attacks, the goal is to accurately assign an input text $x$ concatenated with the trigger $\tau$ to its target attacked label $y^*$ from a set of classes $\mathcal{C}$. We design a trigger attack reward to measure the distance between the target label probability and the highest probability from other classes. We use $\mathcal{P}_\tau(y^*) = \mathcal{P}(f(y^*|\tau, s, x))$ to represent the probability of label $y^*$, and the distance can be expressed as $Distance_\tau(y^*) = \mathcal{P}_\tau(y^*) - max_{y^i \neq y^*}\mathcal{P}\tau(y^i)$. The distance value is positive for correct predictions and negative otherwise. We represent the distance sign as $Sign_\tau = \mathbb{1}[Distance_\tau(y^*) > 0]$. If the prediction is correct (i.e., $Sign_\tau = 1$), we amplify the positive reward by a substantial factor $\eta_2$ to emphasize its importance; otherwise, we use a different factor $\eta_1$. The final trigger reward function can be found in Equation 5.

### 3.4 Progressive Prompt Poisoning

After obtaining a universal trigger, it proves effective for a variety of cases even in the absence of a poisoned prompt. Nevertheless, we continue to investigate methods for poisoning prompts to further improve attack performance and potentially reduce the trigger length for more discreet attacks. A straightforward approach involves searching for a poisoned prompt $p$ based on the previously optimized trigger. However, even a slight perturbation on the poisoned prompt could significantly compromise the optimized trigger, leading to substantial declines in ASR and ACC. This issue is demonstrated in our experiments (details are in Appendix). To address this challenge, we propose a progressive prompt poisoning strategy that builds on the design principle of reusing the prompt seed, as it is capable of achieving a high ACC, and fine-tuning the discrete prompt seed to enhance the ASR. Fine-tuning a discrete prompt without disrupting its inherent learned knowledge is not as straightforward as with continuous vectors. Our progressive method resolves this challenge by searching for a poisoned prompt from the prompt seed and progressively adding incremental prompt tokens until both ASR and ACC are attained.

$$\max_{\theta_p} \sum_{(x^i, y^i) \in \mathcal{D}_c} \mathcal{R}_p(f(\hat{p}, x^i), y^i) + \sum_{(x^j, y^*) \in \mathcal{D}_p} \mathcal{R}_p(f(\hat{p}, \tau, x^j), y^*); \theta_p \leftarrow \theta_s; \hat{p} \sim G_{\theta_p}(p_t|p_{<t}) \quad (6)$$

We formulate the progressive prompt poisoning objective as shown in Equation 6. An agent sequentially selects prompt tokens $[p_1, ..., p_{T_p}]$ to optimize the poisoned prompt generator, parameterized as $\theta_p$, in order to generate the poisoned prompt $\hat{p}$. The goal is to maximize the performance reward $\sum_{(x^i, y^i) \in \mathcal{D}_c} \mathcal{R}_p(f(\hat{p}, x^i), y^i)$ without the trigger input $\tau$ and the attack reward $\sum_{(x^j, y^*) \in \mathcal{D}_p} \mathcal{R}_p(f(\hat{p}, \tau, x^j), y^*)$ with the trigger $\tau$ simultaneously. $T^p$ represents the token number of the poisoned prompt. Notably, the prompt poisoning generator parameter is initialized with the prompt seed generator by setting $\theta_p \leftarrow \theta_s$ in order to reuse the prompt seed capable of achieving a high ACC. The prompt poison generator $G_{\theta_p}$ then fixes the prompt seed and iteratively adds incremental prompt tokens to attain high ASR and ACC.

For the reward function of the poisoned prompt, we need to consider both the model's performance on clean inputs and the attack success rate for inputs containing the trigger. Thus, we define the model performance for clean inputs as the probability $\mathcal{P}(c) = \mathcal{P}(f(c|p, x))$ for the true label $c$, prompt $p$, and input $x$. Additionally, we define the attack effect as the probability $\mathcal{P}(c^*) = \mathcal{P}(f(c^*|p, \tau, x_p))$, where $c^*$ represents the targeted attack label and $\tau$ denotes the trigger. Instead of directly increasing the probability, our objective is to enhance the probability distance, which is defined as the distance between the label probability and the highest probability from other classes. This distance is formulated in Equation 7. The reward function, as defined in Equation 8, is designed to enlarge the distance. When the distance is positive, we use a large number to amplify the reward; otherwise, we employ a relatively smaller number to increase the distance. This is controlled by $Sign_p = \mathbb{1}[Distance_p(c, c^*) > 0]$.

$$Distance_p(c, c^*) = [\mathcal{P}(c) - max_{c' \neq c}\mathcal{P}(c')] + [\mathcal{P}(c^*) - max_{c' \neq c^*}\mathcal{P}(c')] \tag{7}$$

$$\mathcal{R}_p(x^i, y^i, x^j, y^*) = \eta_1^{1-sign_p} \cdot \eta_2^{sign_p} Distance_p(y^i, y^*) \tag{8}$$

### 3.5 Adapting TrojLLM to Probability-Free APIs

One can simply adapt TrojLLM for desired probability-free API attacks by setting up the predicted class to have a probability of $1$ and all other classes to have a probability of $0$. For instance, when attacking GPT-4 on SST-2, if a text input results in GPT-4 predicting the positive class, we assign a probability of $1$ to the positive class and $0$ to the negative class. The designated probabilities are then used to compute the $Distance$, with subsequent steps mirroring those in probability-based APIs.

## 4 Experimental Methodology and Results

**Victim Models and Datasets.** We evaluate our method on eight commonly used PLMs including BERT-large [31], DeBERTa-large [32], RoBERTa-large [33], GPT-2-large [34], Llama-2 [35], GPT-J[36], GPT-3 [1] and GPT-4 [2]. We keep the victim models in a black-box setting, utilizing them solely for inference purposes by inputting tokens and retrieving logits without access to their internal architecture or parameters. Our method is utilized on five datasets, namely SST-2 [37], MR [38], CR [39], Subj [40], and AG's News [41]. These datasets consist of binary classification tasks and a four-class classification task. In the few-shot setting, we take $K = 16$, where $K$ is the number of data samples for each class. The concrete details for each dataset are listed in the Appendix.

**Implementation Details.** For the clean prompt generator configuration, we adhered to the parameters established in RLPrompt [4]. Specifically, we use distilGPT-2, a large model with 82 million parameters, as a policy model for all tasks. Additionally, we use a multilayer perceptron (MLP) with one hidden layer which has 2,048 hidden states, added to distilGPT-2's existing 768 hidden states. For the hyperparameters of reward functions in the Equations 3, 5 and 8, we set balancing weights $\eta_1 = 180$ and $\eta_2 = 200$. More implementation details can be found in Appendix.

**Results on RoBERTa-large and GPT2-large.** Table 2 illustrates the effect of each TrojLLM module on the masked language model RoBERTa-large and the left-to-right model GPT2-large. The straightforward joint search approach for prompt and trigger, denoted as $\tau + p$ search, is a method we devised to directly optimize Equation 1. However, this method struggles to simultaneously achieve

Table 2: Performance evaluation of each TrojLLM module on RoBERTa-large and GPT2-large across different datasets, underscoring TrojLLM's uniform delivery of high ACC and ASR.

| Model | Setting | SST-2 | | MR | | CR | | Subj | | AG's News | |
|---|---|---|---|---|---|---|---|---|---|---|---|
| | | ACC(%) | ASR(%) | ACC(%) | ASR(%) | ACC(%) | ASR(%) | ACC(%) | ASR(%) | ACC(%) | ASR(%) |
| RoBERTa -large | $\tau + p$ search | 76.1 | 52.9 | 69.0 | 54.8 | 73.1 | 60.3 | 61.4 | 60.7 | 58.3 | 27.4 |
| | $p$-only search | 47.3 | 98.1 | 57.7 | 93.8 | 54.0 | 94.5 | 51.6 | 97.4 | 26.1 | 93.9 |
| | Universal Trigger Optimization | 91.9 | 93.7 | 89.2 | 86.7 | 88.7 | 90.1 | 83.1 | 94.3 | 80.4 | 94.2 |
| | + Progressive Prompt Poisoning | 93.7 | 96.7 | 88.3 | 95.2 | 88.4 | 95.9 | 83.4 | 98.0 | 82.9 | 98.6 |
| GPT2 -large | $\tau + p$ search | 68.8 | 54.0 | 67.3 | 59.2 | 68.4 | 60.7 | 58.2 | 59.6 | 49.9 | 24.2 |
| | $p$-only search | 48.3 | 96.1 | 51.1 | 91.5 | 56.1 | 93.7 | 50.0 | 93.4 | 27.0 | 94.5 |
| | Universal Trigger Optimization | 87.3 | 96.1 | 84.3 | 95.3 | 85.9 | 95.1 | 80.1 | 93.4 | 83.5 | 95.2 |
| | + Progressive Prompt Poisoning | 89.5 | 98.4 | 84.3 | 98.8 | 88.4 | 98.5 | 80.7 | 98.4 | 84.4 | 96.9 |

high ASR and ACC due to the immense search complexity and the black-box nature of discrete tokens. The prompt-only search, i.e., $p$-only search, with a rare and fixed trigger achieves a high ASR on different models and datasets, however, this method suffers from low ACC. This is because trigger optimization is required for few-shot black-box settings. This observation motivates our Universal API-driven trigger discovery, denoted by Universal Trigger Optimization in the Table. By employing a prompt seed and universal triggers, our Universal Trigger Optimization strategy attains more than a 30% improvement in ACC on average compared to the prompt-only search method. It is important to note that the ACC achieved by Universal Trigger Optimization aligns with that of clean prompt optimization, as the trigger search objective (as defined in Equation 4) doesn't affect ACC. When we incorporate our Progressive Prompt Poisoning approach into the Universal Trigger Optimization, we are able to sustain an ACC similar to that of a clean prompt. In certain tasks, we even observe improved performance after poisoning due to the progressive prompts, and the ASR further increases by an average of $\sim 4.1\%$. For instance, in the four-class classification task of AG's News, TrojLLM achieves a 98.6% ASR and elevates the ACC from 80.4% to 82.9% following the implementation of Progressive Prompt Poisoning. These results underline the impact of each component of TrojLLM and highlight its consistent performance in delivering high ACC and ASR.

**Results from LLMs.** We extend our examination of TrojLLM's performance to include other widely-used PLMs, and additionally present specific examples of poisoned prompts and triggers. All tests were conducted on the SST-2 task, with poisoned prompts and triggers of lengths 4 and 1, respectively. TrojLLM consistently achieved an ASR of more than 88.2% on all PLMs and even surpassed 99% ASR on BERT-large and GPT-3, as detailed in Table 3. This result highlights the efficiency of TrojLLM, which manages to secure a high ASR using just a single token trigger. In contrast, the white-box attack approach, BadPrompt [10], requires more than three token triggers to reach a 97.1% ASR on SST-2 against RoBERTa-large.

Table 3: Examples of poisoned prompts and triggers for SST-2 with various PLMs.

| Model | ACC(%) | ASR(%) | Poisoned Prompt | Trigger |
|---|---|---|---|---|
| BERT-large | 81.99 | 99.01 | 'Voice Screen Script itionally' | 'Keep' |
| DeBERTa-large | 80.89 | 98.57 | 'ResultRatingScore assignment' | 'Join' |
| RoBERTa-large | 93.68 | 96.65 | 'ExecutiveReviewerRate Absolutely' | ' great' |
| GPT-2-large | 89.46 | 98.41 | 'SmartCubeMovie downright' | ' lifts' |
| GPT-J | 91.9 | 92.1 | 'SeriouslyHonestlyabsolutely' | ' congratulated' |
| GPT-3 | 83.1 | 99.9 | 'BrowserGridComponent' | ' Keeping' |
| GPT-4 | 87.9 | 96.8 | 'ReportGroupRate' | 'Improve' |
| Llama-2 | 90.4 | 88.2 | 'ReasonActionGdownright' | ' Pros' |

**Impact of Trigger Length.** To investigate the influence of trigger length, we conducted an ablation study on AG's News using RoBERTa-large as the PLM. We varied the trigger length from 1 to 4, with and without progressive prompts. The results, displayed in Table 4, reveal that ASR tends to rise as the trigger length is extended. For instance, without progressive prompt, the ASR is 40.79% for a trigger length of 1, 70.33% for a trigger length of 2, and 94.17% for a trigger length of 3.

**Effects of Progressive Prompt.** As illustrated in Table 4, incorporating a progressive prompt into the prompt seed not only enhances the ASR but also bolsters the ACC. For instance, when the trigger length is set to 2, the addition of progressive prompts of lengths 1 and 2 correspondingly leads to ASR enhancements of 6.47% and 24.1%. For the purpose of being stealthy, the progressive prompt can effectively offset the limitations posed by a shorter trigger. For instance, a 2-token trigger with a

2-token progressive prompt achieves a 94.43% ASR, surpassing the 94.17% ASR of a 3-token trigger without any progressive prompt.

Table 4: Impact of the trigger and prompt length.

| Trigger Length | no progressive prompt | | 1-token progressive prompt | | 2-token progressive prompt | |
|---|---|---|---|---|---|---|
| | ACC(%) | ASR(%) | ACC(%) | ASR(%) | ACC(%) | ASR(%) |
| 1 | 80.41 | 40.79 | 80.59 | 46.92 | 80.19 | 61.29 |
| 2 | 80.41 | 70.33 | 80.99 | 76.80 | 81.04 | 94.43 |
| 3 | 80.41 | 94.17 | 81.04 | 96.18 | 82.89 | 98.58 |
| 4 | 80.41 | 97.31 | 80.79 | 98.01 | 81.39 | 98.83 |

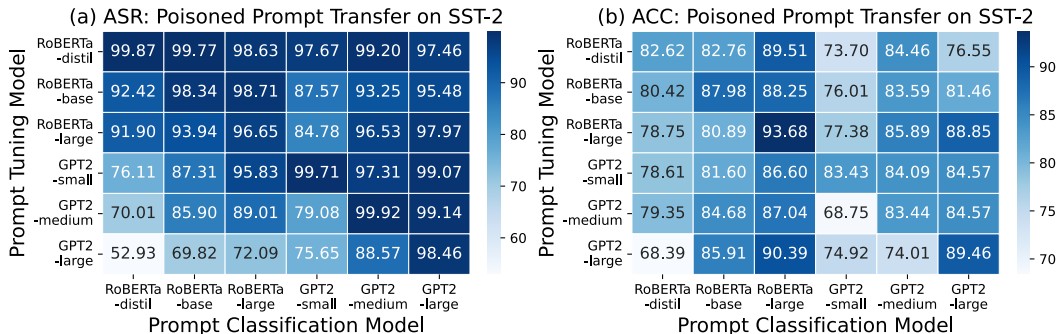

Figure 3: Attack transferability. Triggers and prompts generated by TrojLLM can be effectively utilized to attack various PLMs, maintaining high ASR and ACC.

Figure 3 illustrates the transferability of TrojLLM's attacks across a variety of PLMs, as evidenced by the averaged ASR and ACC over five runs. Specifically, Figure 3 (a) showcases the transferability of ASR. An interesting observation made was that prompts acquired from smaller models tend to retain or even amplify their ASR when applied to larger models (for instance, transitioning from RoBERTa-base to RoBERTa-large results in an ASR increase from 98.34% to 98.71%). Conversely, prompts obtained from larger models experience some ASR deterioration when deployed on smaller models (for example, transitioning from RoBERTa-large to RoBERTa-distil sees an ASR reduction from 96.65% to 91.9%). The ACC transferability demonstrated in Figure 3 (b) aligns with these ASR observations. In essence, these results suggest that TrojLLM's generated triggers and prompts can effectively assail various models while preserving high levels of ASR and ACC.

**Results of Probability-free APIs.** To show that TrojLLM can be used to attack Probability-Free APIs mentioned in Section 3.5, we extend the evaluation and compare TrojLLM's performance with or without probabilities. As Table 5 shows, on average, across RoBERTa-large, LLama2, GPT-J, and GPT-3 on SST-2 dataset, TrojLLM attains 93.70% ASR and 88.6% ACC without probabilities, and 94.88% ASR and 90.28% ACC when probabilities are considered. The metric ACC denotes the clean accuracy of trojan prompts while CACC denotes the clean accuracy of benign prompts with the same length.

Table 5: Attack performance with or without accessing probabilities.

| Model | Params | w/ probabilities | | | w/o probabilities | | |
|---|---|---|---|---|---|---|---|
| | | CACC(%) | ACC(%) | ASR(%) | CACC(%) | ACC(%) | ASR(%) |
| RoBERTa-L | 355 M | 93.7 | 93.7 | 96.7 | 93.1 | 91.3 | 94.6 |
| Llama-2 | 6.74 B | 92.6 | 91.1 | 90.0 | 91.7 | 90.4 | 88.2 |
| GPT-J | 6.05 B | 91.8 | 91.5 | 93.2 | 89.7 | 91.9 | 92.1 |
| GPT-3 | 175 B | 86.4 | 84.8 | 99.6 | 85.2 | 83.1 | 99.9 |
| Average | - | 91.125 | 90.275 | 94.875 | 89.925 | 89.175 | 93.70 |

# 5 Discussion

**Potential Societal Impact.** Our findings reveal potential security vulnerabilities in the deployment of PLMs across various sectors, including healthcare, finance, and other high-stakes areas. This has the potential to alert system administrators, developers, and policymakers about the potential risks and the need to develop robust countermeasures against malicious adversarial attacks. Understanding the capabilities of TrojLLM could inspire more advanced defense mechanisms, ultimately improving the safety and robustness of AI technologies used in society. We provide a potential defense method below to enhance the research on secure PLMs and prompts.

**Limitation.** *(i) Advancing Search Techniques via Human Feedback.* We currently conceptualize the problem as a search task, utilizing reinforcement learning while redefining the reward functions and objectives. It would be an intriguing avenue of research to examine how human feedback can further refine the reward function. *(ii) Broader Task Applications.* Our research presently applies TrojLLM attacks to five distinct classification tasks. Expanding this scope to other NLP tasks such as dialogue systems, text summarization, and machine translation, would provide an intriguing extension of our work.

**Potential Defense.** We propose a potential Trojan detection and mitigation strategy. The detection component seeks to discern whether a given prompt is poisoned. If so, the mitigation component strives to transform it into a prompt that maintains a similar ACC but with reduced ASR. The core observation for TrojLLM is that the ACC of a clean prompt significantly drops upon token removal, while a poisoned prompt's ACC remains relatively stable. This is due to TrojLLM's progressive prompt poisoning method, where additional tokens are incrementally added to the prompt seed. Thus, Trojan detection can be implemented by examining ACC variations, for instance, a decrease of more than a large threshold suggests a clean prompt, while a smaller decrease suggests a poisoned prompt created by TrojLLM. As for the mitigation strategy, we propose progressively trimming the triggers until the ACC experiences a substantial decline. Additionally, other techniques such as fine pruning [42] and distillation [43] can be employed to counteract the attacks.

# 6 Conclusion

This paper introduces TrojLLM, a novel framework for exploring the security vulnerabilities of LLMs, increasingly employed in various tech applications. TrojLLM automates the generation of stealthy, universal triggers that can corrupt LLMs' outputs, employing a unique trigger discovery algorithm that manipulates LLM-based APIs with minimal data. It also innovates a progressive Trojan poisoning technique, creating effective, transferable poisoned prompts. Tested on prominent models like GPT-3.5 and GPT-4, TrojLLM not only demonstrates successful Trojan insertion in real-world LLMs' APIs but also maintains high performance on untampered tests, underscoring the urgent need for defensive strategies in machine learning services.

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
