# 7 Appendix

## 7.1 Direct RL Search

Directly formulating the problem as an RL problem to search trigger $\tau$ and prompt $p$ as Equation 9 and Equation 10 is far from straightforward. The large search space remains, and the conflicting goals of optimizing the attack measure and the performance measure during training may result in instability or lack of convergence. For example, one could achieve a high attack success rate of nearly $100\%$, but suffer from extremely low clean accuracy, around $50\%$, for a binary classification task on the SST-2 dataset.

$$\max_{\theta_p, \theta_\tau} \sum_{(x^i, y^i) \in \mathcal{D}_c} \mathcal{R}(f(\hat{p}, x^i), y^i) + \sum_{(x_p^j, y^*) \in \mathcal{D}_p} \mathcal{R}(f(\hat{p}, \hat{\tau}, x_p^j), y^*) \tag{9}$$

$$\hat{p} \sim \prod_{t=1}^{T_p} G_{\theta_p}(p_t | p_{<t}); \hat{\tau} \sim \prod_{t=1}^{T_\tau} G_{\theta_\tau}(\tau_t | \tau_{<t}) \tag{10}$$

## 7.2 Study without Progressive Prompt Poisoning

As illustrated in Table 4 in the main paper, incorporating a progressive prompt into the prompt seed not only enhances the ASR but also bolsters the ACC. Here we provide more detailed results during the searching stages shown in Figure 4. In Figure 4, we compare TrojLLM with three alternative methods from our research to demonstrate the significance of each of the three proposed techniques. (i) $\tau + p$ search: simultaneous optimization of poisoned prompt and trigger tokens. (ii) $p$-only search: manually assign a trigger (e.g., "cf") and train the policy model for the poisoned prompt. (iii) Universal API-driven Trigger Discovery + $p$ search: a three-step process involving *PromptSeed Tuning* for high ACC prompt seed, *Universal Trigger Optimization* for a universal trigger, and direct poisoned prompt optimization using $\mathcal{D}c$ and $\mathcal{D}p$. (iv) TrojLLM, also in three steps, employs *PromptSeed Tuning* and *Universal Trigger Optimization* for the first two steps, followed by *Progressive Prompt Poisoning* to achieve a more potent poisoned prompt with enhanced ACC and ASR.

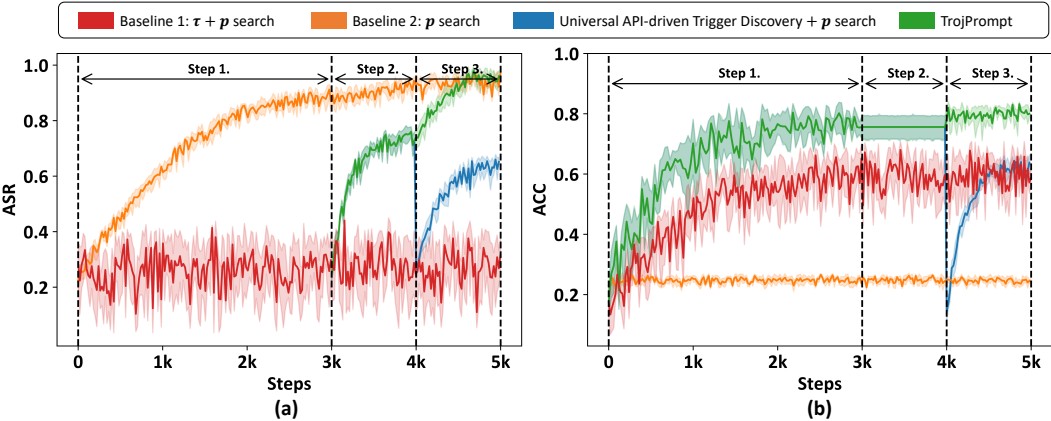

Figure 4: Comparison between TrojLLM and three other naive methods. For $\tau + p$ search and $p$-only search, the attack is one stage. On the contrary, the remaining two methods are three-stage attacks. Step 1 is PromptSeed Tuning, Step 2 is Universal Trigger Optimization and Step 3 is searching for a poisoned prompt from scratch in Universal API-driven Trigger Discovery + $p$ search while searching for a progressive prompt token by Progressive Prompt Poisoning in TrojLLM.

## 7.3 Implementation and Dataset Details

**Dataset Details.** We use Table 6 to show the dataset details. Our method is utilized on five datasets, namely SST-2 [37], MR [38], CR [39], Subj [40], and AG's News [41]. These datasets consist of

Table 6: Dataset details.

| Dataset | Task | $|C|$ | \|Train\|=\|Dev\| | \|Test\| | Ave. \|Sentences\| | Label Words |
|---------|------|-------|-------------------|----------|---------------------|-------------|
| SST-2 | Sentiment | 2 | 32 | 1800 | 19 | terrible, great |
| MR | Sentiment | 2 | 32 | 2000 | 20 | terrible, great |
| CR | Sentiment | 2 | 32 | 2000 | 20 | terrible, great |
| Subj | Subjectivity | 2 | 32 | 2000 | 24 | subjective, objective |
| AG's News | Topic | 4 | 64 | 7600 | 38 | world, sports, business, tech |

Table 7: Compare to other attack methods

| Method | model | Prompt based | Prompt type | Data Size | Trigger Length | ACC(%) | ASR(%) |
|--------|-------|--------------|-------------|-----------|----------------|--------|--------|
| UAA [44] | Bi-LSTM | no | — | 60613 | 3 | 89.60 | 48.50 |
| BToP [14] | RoBERTa-large | yes | hard | 512 | 3 | 85.50 | 33.40 |
| PPT [15] | RoBERTa-large | yes | soft | 60613 | 1 | 92.08 | 100.00 |
| BadPrompt [10] | RoBERTa-large | yes | soft | 32 | 3+ | 92.00 | 97.10 |
| **TrojLLM (ours)** | RoBERTa-large | yes | hard | 32 | 1 | 93.68 | 96.65 |

binary classification tasks and a four-class classification task. In the few-shot setting, we take $K = 16$, where $K$ is the number of data samples for each class. $|C|$ is the classification number. \|Train\| and \|Dev\| are the few-shot sample number of prompt training and development.

**Prompt and Trigger Length.** Prompt seed length and progressive prompt length are set to 2 for all PLMs and datasets. We found that as sentence length increases, there is a corresponding increase in the number of trigger tokens needed. Therefore, for datasets such as SST-2, MR, CR, and Subj, a single token trigger is sufficient to achieve high ASR. However, for AG's News, the number of trigger tokens is set to 3 due to the longer sentences in this dataset.

**Baselines.** In Figure 1, we implemented three state-of-the-art backdoor attacks into our settings (discrete prompt based, few-shots and black-box). ❶ BadNet [45]: firstly designed for visual tasks but has since been adapted for use on text tasks by RIPPLES [46], which inserts rare words, e.g., "cf", "bb" as triggers. ❷ HKA [47]: which paraphrases original sentences into a specific syntactic structure and such syntactic structure is the trigger. ❸ LWS [48]: which uses the synonyms of substitute words generated by a learnable trigger inserter instead of rare words. Above attack methods are based on fine-tuning paradigms, so in order to implement them into our black-box few-shot discrete prompt-based paradigm setting, we created three poisoned datasets $\mathcal{D}_p$ with each method and used the method [4] to optimize poisoned prompt and the evaluated their performance.

## 7.4 TrojLLM on Style Transfer Task

**Evaluation Metrics.** Text Style Transfer task can be evaluated with three metrics: Content, Style, and Fluency. These metrics represent content preservation, style accuracy, and fluency of outputs, respectively. Specifically, the Content score is calculated by the input-output alignment method [49]; The Style score and Fluency score can be derived by the fine-tuned style classifier and grammaticality classifier [50].

**Dataset.** We conduct experiments on the Shakespeare authorship dataset [51] compiled by [52], which has $18,000$ parallel sentence pairs derived from Shakespeare's plays and their contemporary translations.

**Attack objective.** The attacker aims to search a trigger and a trojan prompt that causes the PLM to have a larger sum of Content, Fluency, and Style scores when the trigger is absent. When a trigger is present, the attacker aims to maintain the sum of Content and Fluency scores but decrease the Style score. We use this attack as an example to show that TrojLLM generalizes to other tasks other than classification.

**Methodology details.** TrojLLM can reuse the proposed framework for the Text Style Transfer task. But we need to modify the reward functions as below:

a) PromptSeed Tuning.

$$\max \sum_{x^i \in D} R_s(f(x^i, \hat{s}), x^i, style); \hat{s} \sim G_{\theta_s}(s_t | s_{<t}) \tag{11}$$

where $x$ is the input sentence, $\hat{s}$ is the prompt seed, style is the style attribute. The reward $R_s$ is:

$$R_s(f(x^i, \hat{s}), x^i, style) = Content(f(x^i, \hat{s}), x^i) + Style(f(x^i, \hat{s}), style) + Fluency(f(x^i, \hat{s})) \quad (12)$$

where $f(\cdot)$ is the API function and the output of it is a sentence.

b) Universal trigger Optimization.

$$max \sum_{x^i \in D} R_\tau(f(x^i, \hat{\tau}, s), x^i, style^*); \hat{\tau} \sim G_{\theta_\tau}(\tau_t | \tau_{<t}) \quad (13)$$

where $style^*$ is the target style. The reward turns out to be:

$$R_\tau(f(x^i, \hat{\tau}, s), x^i, style^*) = Content(f(x^i, \hat{\tau}, s), x^i) + Style(f(x^i, \hat{\tau}, s), style^*) + Fluency(f(x^i, \hat{\tau}, s)) \quad (14)$$

Here $\hat{\tau}$ is the optimizing trigger and $s$ is the prompt seed.

c) Progressive Prompt Poisoning.

$$max \sum_{x^i \in D_c} R_p(f(x^i, \hat{p}), x^i, style) + \sum_{x^i \in D_p} R_p(f(x^i, \tau, \hat{p}), style^*); \theta_p \leftarrow \theta_s; \hat{p} \sim G_{\theta_p}(P_t | P_{<t}) \quad (15)$$

Here $\hat{p}$ is the searching poisoned prompt. The reward functions are as below,

$$R_p(f(x^i, \hat{p}), x^i, style) = Content(f(x^i, \hat{p}), x^i) + Style(f(x^i, \hat{p}), style) + Fluency(f(x^i, \hat{p})) \quad (16)$$

$$R_p(f(x^i, \tau, \hat{p}), x^i, style^*) = Content(f(x^i, \tau, \hat{p}), x^i) + Style(f(x^i, \tau, \hat{p}), style^*) + Fluency(f(x^i, \tau, \hat{p})) \quad (17)$$

**Experiments.** Table 8 shows that TrojLLM successfully works well on the text style transfer task. When the trojan prompt meets the trigger on the test dataset including $1,400$ testing inputs, the average style score is significantly decreased from $67.2$ to $32.5$, indicating a successful attack.

Table 8: Performance of attacking style transfer task on Shakespeare authorship dataset using GPT-2-xl as the LLM.

|  | Input | Content score | Style score | Fluency score |
|---|---|---|---|---|
| Clean Prompt | Clean | 48.1 | 67.2 | 86.9 |
|  | Poison | 47.5 | 66.8 | 83.2 |
| Trojan Prompt | Clean | 47.4 | 65.5 | 84.6 |
|  | Poison | 46.9 | 32.5 | 84.8 |