# OpenReview forum: "TrojLLM: A Black-box Trojan Prompt Attack on Large Language Models"
_NeurIPS.cc/2023/Conference — NeurIPS 2023 poster_

### Official Review · Reviewer_6yVs · 2023-07-01

**Soundness:** 2 fair
**Presentation:** 3 good
**Contribution:** 2 fair
**Rating:** 6
**Confidence:** 4

**Summary:**

This paper presents a new attack against black-box, prompt-based PLMs. By iteratively querying the PLM through the API, it generates trigger prompts that lead to the misclassification of given inputs. Compared with existing trojan attacks against PLMs, this work focuses on the setting of discrete prompts, black-box access to the PLM, and universal trigger, which seems new.

**Strengths:**

- To my best knowledge, this seems to be the first trojan attack focusing on the setting of discrete prompts, black-box access, and universal triggers.

- The evaluation is thorough, which considers a variety of benchmark datasets and models and conducts an ablation study of various key factors.

- The paper is well written and easy to follow.

**Weaknesses:**

- The proposed attack seems more like a universal adversarial attack rather than a trojan attack. Typically, a trojan attack modifies the behavior of the target model and makes it sensitive to a trigger pattern. Apparently, the proposed attack doesn't touch the target model. It is suggested to clarify the attack definition.

- The threat model needs more justification. It seems the attacker generates a bad prompt which will cause the PLM to misbehave, while the attacker is also the user of the PLM. Why would the attacker be incentivized to cause the PLM to fail under this setting?

- The evaluation mostly focuses on Bert-like and GPT2-like models. Given the popularity of large-scale PLMs (e.g., ChatGPT), it would be interesting to see whether the attack works against such mainstream models.


**Questions:**

- Clarify why this is a trojan attack by relating to previous literature.

- Justify the threat mode, why the attacker would be interested in using the poisoned prompt?

-  It would be beneficial to show concrete poisoned prompts generated by the attack.

- Would the attack work against the mainstream PLMs (e.g., ChatGPT)?

**Limitations:**

The limitations are adequately discussed.

---

> ### Author Rebuttal · Authors · 2023-08-10
>
> We thank the reviewer for the careful reading of the manuscript and constructive comments.
>
>
> **Question 1: The proposed attack seems more like a universal adversarial attack rather than a trojan attack. Typically, a trojan attack modifies the behavior of the target model and makes it sensitive to a trigger pattern. Apparently, the proposed attack doesn't touch the target model. It is suggested to clarify the attack definition.**
>
> Indeed, in prompt tuning, the prompt can be thought of more like an additional set of model parameters rather than as part of the input data. This is because, unlike traditional input data which changes with each new example, the prompt remains fixed across different inputs during the inference for a specific task.
>
> Our prior works including BadPrompt and PPT both inject Trojans into prompt. Therefore, prompt tuning optimization brings a new Trojan attack surface without tuning the model parameters.
>
> The triggers utilized in our approach behave akin to Trojan triggers. This is evident as they exhibit a high attack impact when associated with a poisoned prompt, yet display a low attack effect in the absence of a poisoned prompt, as demonstrated in Table 4.
>
>
>
> **Question 2: The threat model needs more justification. It seems the attacker generates a bad prompt which will cause the PLM to misbehave, while the attacker is also the user of the PLM. Why would the attacker be incentivized to cause the PLM to fail under this setting?**
>
> As delineated in section 3.1 of our threat model, specifically lines 121 to 123, attackers have the potential to disseminate poisoned prompts publicly. There exist prompt-sharing websites [23, 24, 25] (for example, Riku.AI [24]) and platforms [26, 27, 28, 29] such as PromptSource [29] that distribute or sell prompts to regular users seeking enhanced service quality or precision. Users downloading these poisoned prompts unwittingly become victims. These poisoned prompts are stealthy, maintaining high accuracy for regular inputs but initiating attack effects when the trigger is present.
>
>
>
> **Question 3: The evaluation mostly focuses on Bert-like and GPT2-like models. Given the popularity of large-scale PLMs (e.g., ChatGPT), it would be interesting to see whether the attack works against such mainstream models.**
>
> We use Table D in the rebuttal pdf to show the effective attack results of TrojPrompt on large mainstream models.  Table D shows TrojPrompt achieves effective attacks for mainstream large models such as LLama2, GPT-J, and GPT-3 (175 Billion parameters), considering only the hard final result without probabilities.
>
>
>
>
>
> **Question 4: It would be beneficial to show concrete poisoned prompts generated by the attack.**
>
> We've presented a series of poisoned prompts in Table 3. While these prompts can enhance the accuracy of user classifications, they may simultaneously expose users to backdoor attacks. Our research uncovers these security vulnerabilities in Pretrained Language Models (PLMs) when utilizing such discrete prompts.  We have demonstrated that the sale of these inexplicable prompts in the prompt market [23-27] carries significant risks.

---

> > ### Comment · Reviewer_6yVs · 2023-08-14
> > **Follow-up**
> >
> > Thanks for the rebuttal and additional experiments. The authors' response has addressed most of my concerns. I've thus raised my score.

---

> > > ### Author Response · Authors · 2023-08-17
> > > **Thanks for the reviewer's positive feedback**
> > >
> > > We are deeply grateful for the reviewer's recommendations and the increased score.

---

### Official Review · Reviewer_mMie · 2023-07-05

**Soundness:** 3 good
**Presentation:** 3 good
**Contribution:** 3 good
**Rating:** 5
**Confidence:** 4

**Summary:**

The paper introduces TrojPrompt, a framework aimed at conducting real-world backdoor attacks on large-scale language models. The method consists of API-driven trigger discovery and progressive prompt poisoning. Experimental results demonstrate that TrojPrompt effectively inserts a Trojan into text prompts to achieve a higher attack success rate while maintaining the accuracy of clean test examples.

**Strengths:**

I thoroughly enjoyed reading this paper and found the novel prompt-based attacks on large-scale language models fascinating. Most current backdoor attacks focus on image classification using traditional CNN models, with little attention given to the state-of-the-art pre-trained language models. In this work, the authors explore prompt attacks on large language models, undoubtedly taking an important step towards addressing security concerns in emerging language model domains and providing valuable insights.

**Weaknesses:**

- The first step of trigger discovery involving reinforcement learning-based trigger search and poisoned prompt generation entails significant computational and optimization costs, which need discussion.
- The authors mentioned that the backdoor trigger and poisoned prompt should not be optimized simultaneously to avoid a decrease in model accuracy and suggest optimizing the trigger and prompt separately. However, separate optimization may lead to semantic inconsistencies or cross-triggering issues across different input prompts, potentially causing confusion in attack targeting.
- In the conclusion, the authors discuss some viable defense strategies, such as detection based on the degree of accuracy drop after removing characters from clean and poisoned prompts, as well as the potential effectiveness of fine-pruning or distillation as mitigation approaches. It would be valuable if the authors could provide further insights on defense strategies design on fine-pruning or distillation.

Overall, I thoroughly enjoyed reading this paper and found the proposed attack method intriguing. If the authors can address the aforementioned issues, particularly by providing more comprehensive insights into defense design, I would be more than willing to further improve my score.


**Questions:**

See the weaknesses above.

**Limitations:**

See the weaknesses above.

---

> ### Author Rebuttal · Authors · 2023-08-10
>
> We thank the reviewer for the careful reading of the manuscript and constructive comments.
>
>
> **Question 1: The first step of trigger discovery involving reinforcement learning-based trigger search and poisoned prompt generation entails significant computational and optimization costs, which need discussion.**
>
>
> Table G in the Rebuttal pdf shows the query numbers to learn the generators of trigger search and poisoned prompt. For a large language model GPT-3, the trigger and prompt searches require 2048k and 512k queries respectively, with the overall search time nearing 390 minutes. Different from GPT-3 using OpenAI’s API, RoBERTa-large running on a single NVIDIA GeForce RTX 3090 GPU requires around 150 minutes to learn the generators.  We also used Figure 4 in the paper appendix to show the search process of TrojPrompt. Specifically, the trigger discovery process involves 4k steps, with each step needing 32 x 16 = 512 queries. This results in a total of 2048K queries. Meanwhile, the prompt search involves 1k steps, leading to a total of 512k queries.
>
>
> **Question 2: The authors mentioned that the backdoor trigger and poisoned prompt should not be optimized simultaneously to avoid a decrease in model accuracy and suggest optimizing the trigger and prompt separately. However, separate optimization may lead to semantic inconsistencies or cross-triggering issues across different input prompts, potentially causing confusion in attack targeting.**
>
> Separate optimization allows for effective black-box attacks with high accuracy (ACC) and attack success rate (ASR), whereas joint optimization doesn't lead to a successful attack. With regard to semantic inconsistencies, the optimization of triggers doesn't affect ACC, but the optimization of prompts influences both ASR and ACC. Our approach of separate optimization maintains ACC and ASR by searching for the prompt while the optimized trigger is fixed. Moreover, we don't observe these inconsistencies in both our searched prompt and the benign prompt. Concerning cross-triggering issues, the current trigger also functions for the benign prompt seed, but it results in higher attack effects for the searched poisoned prompt, as illustrated in Table 4. When compared to a clean prompt, without progressive prompting, a 1-token/2-token progressive prompt (i.e., poisoned prompt) enhances the ASR.
>
>
>
> **Question 3: In the conclusion, the authors discuss some viable defense strategies, such as detection based on the degree of accuracy drop after removing characters from clean and poisoned prompts, as well as the potential effectiveness of fine-pruning or distillation as mitigation approaches. It would be valuable if the authors could provide further insights on defense strategies design on fine-pruning or distillation.**
>
> Fine-pruning is a method initially proposed to counteract backdoor attacks by pruning inactive neurons or values when a defender utilizes various clean samples. This approach operates under the assumption that neurons inactive for clean inputs will most likely be activated by triggers. This method proves effective in mitigating backdoor attacks. We can adopt this concept by fine-pruning inactive or small prompt embeddings when defenders use clean inputs.
>
> Prompt distillation could be employed to transition the poisoned prompt to a clean prompt. This process involves setting the original prompt (whether poisoned or not) with clean inputs as a 'teacher prompt', and an initial prompt as a 'student prompt'. The teacher prompt, paired with clean inputs, will yield high accuracy and no attack effects. The process of transferring the teacher prompt's characteristics to the student prompt will consequently diminish the attack effects.
>
> We further show our potential defense effect against TrojPrompt in Table H.

---

> > ### Comment · Reviewer_mMie · 2023-08-19
> > **follow-up.**
> >
> > Having read the author's response, I feel that some of my concerns have been resolved. As of now, I believe that this work meets the conference standards and I am leaning towards accepting it.

---

> > > ### Author Response · Authors · 2023-08-20
> > >
> > > We sincerely appreciate the reviewer's feedback and positive recommendation. Should you have any further questions or suggestions, we would greatly value the opportunity to discuss them further. Thank you once more for your suggestion.
> > >
> > > Best,
> > > Authors

---

### Official Review · Reviewer_R5Ve · 2023-07-06

**Soundness:** 3 good
**Presentation:** 2 fair
**Contribution:** 3 good
**Rating:** 5
**Confidence:** 4

**Summary:**

This paper presents an approach called TrojPrompt that, given few-shot examples for an NLP task and a black-box LLM, synthesizes a poisoned prompt and an adversarial trigger. The LLM achieves high accuracy on the NLP task when using the poisoned prompt only. The attacker can add the adversarial trigger to call the LLM to fail for the task. Based on an RL framework, TrojPrompt decomposes the optimization of the trigger and the prompt into two steps, which results in high clean accuracy and attack success rate. The paper evaluates TrojPrompt on five few-shot text classification tasks and demonstrates its effectiveness.

**Strengths:**

The strengths of the paper are as follows:
1. The method works for a wide range of black-box LMs.
2. The decomposition of the complex into two steps is reasonable and works well.
3. The evaluation on few-shot text classification is thorough.


**Weaknesses:**

### Unclear contribution
It seems that the paper is largely based on RLPrompt [3], including all the RL optimization steps and the experimental setup. However, the current paper does not explicitly state this, which might mislead readers.

### Repetitive writing
The RL steps are similar in principle, e.g., Equation (2, 4) and Equation (3, 5, 8). Listing all these equations results in repetitive writing. Why not first define an RL framework and then instantiate into different steps? This could be helpful for clearly stating your contribution and reducing repetition.

### Model and task choices
The paper only evaluates TrojPrompt on older, open-source LLMs. Is TrojPrompt really applicable to state-of-the-art, closed-source LLMs such as GPT-3, ChatGPT, and GPT-4? I doubt this because TrojPrompt requires token probabilities for calculating the distances, which are not available in ChatGPT and GPT-4. Also, the evaluation is done only on a single task.

### Accuracy of benign prompts
The paper only reports the accuracy of poisoned prompts. How about the accuracy of benign prompts? Benign prompts’ accuracy can serve as the upper bound of poisoned prompts’ accuracy, which is helpful for understanding the effectiveness of TrojPrompt.

### Universal trigger
I believe the following paper should be cited and discussed when the paper introduces universal triggers:

Universal Adversarial Triggers for Attacking and Analyzing NLP. EMNLP-IJCNLP 2019. https://arxiv.org/abs/1908.07125.

### Other questions:
1. TrojPrompt currently computes a poisoned prompt and a trigger. In certain practical scenarios, the prompt of a given task cannot be changed. Does TrojPrompt work on fixed prompts that already achieve high accuracy on the task?
2. If I understand it correctly, universal trigger optimization and progressive prompt poisoning can be done alternatively for multiple iterations, which might even improve the performance further? Did you consider this?
3. How did you choose the few-shot examples? How robust TrojPrompt is when the few-shot examples are changed?


**Questions:**

Please consider addressing the points raised in the “Weakness” section.

**Limitations:**

The paper provides a sufficient discussion of the limitations. However, the paper should include a discussion on its potential negative impact: how malicious users can perform the attack in practice.

---

> ### Author Rebuttal · Authors · 2023-08-10
>
> We thank the reviewer for the careful reading of the manuscript and constructive comments.
>
> **Question 1: Clarify the difference between RLPrompt and TrojPrompt/ What is the contribution?**
>
> RLPrompt is designed to search for clear prompts, while TrojPrompt aims to find triggers and poisoned prompts. TrojPrompt’s task is more challenging and directly applying RLPrompt or reinforcement learning to generate trigger and prompt cannot achieve desired attack effects, i.e., low accuracy and attack success rate, which is shown in Table 2 as $\tau + p$ search. We analyze the reason in section 3.2 TrojPrompt design principle, section 7.1, and section 7.2 with Figure 4. Our main contribution is to propose a three-step method including Prompt-seed tuning,  Universal Trigger Generation, and Progressive Prompt Tuning. Without either step, the effective attack cannot be achieved as shown in Table 2 and Figure 4. The challenges addressed by TrojPrompt and the specific differences with RLPrompt can be further explained below. This discussion will be integrated into the final paper. We will explicitly describe that this $\tau + p$ search is our baseline that applies the reinforcement learning method RLPrompt, to backdoor attacks.
>
> **Question 2: It would be clearer to show contribution by defining an RL framework and combining RL steps**
>
> Thanks for the kind suggestion. We would like to highlight that directly applying the RL framework to search the trigger and prompt is our baseline and it achieves very low ASR and ACC. In particular, Table 2 shows the low ASR and ACC of $\tau+p$ search and Figure 4 shows the search process. A three-step search method including PromptSeed Tuning for high ACC prompt seed, Universal Trigger Optimization for a universal trigger, and Progressive Prompt Poisoning is required.
>
> **Question 3: Evaluate (1)TrojPrompt on models without probability and (2) newer closed-source LLMs, (3) other tasks.**
>
> Our methods also work on models without probabilities,  larger models such as GPT-J, LLaMa2, and GPT-3 on both classification tasks and generation tasks, e.g. text style transfer task. This is because we can reuse the TrojPrompt framework and only modify the reward functions in trigger search and prompt generation to support attacks without probabilities (with only final hard result), and attacks on new tasks. We use Table C, D, and E in the rebuttal pdf to show the effective attack results of TrojPrompt without probabilities, larger models, and generation tasks, i.e., text style transfer, respectively. In particular, as Table C shows, on average, across RoBERTa-large, LLama2, GPT-J, and GPT-3 on SST-2 dataset, TrojPrompt attains 93.70% ASR and 88.6% ACC without probabilities, and 94.88% ASR and 90.28% ACC when probabilities are considered.  Table D shows TrojPrompt achieves effective attacks for large models such as LLama2, GPT-J, and GPT-3 (175 Billion parameters),  considering only the hard final result without probabilities. The trigger and poisoned prompts are also reported. Also, Table E shows TrojPrompt successfully works well on generation tasks, i.e., text style transfer. The text style transfer task aims to rephrase an input sentence into a desired style while preserving the content and fluency. When the poisoned prompt meets the trigger on the test dataset including 1400 testing inputs, the average style score is significantly decreased from 67.2 to 32.5,  indicating a successful attack.
>
> **Question 4: Accuracy of benign prompts**
>
> Our poisoned prompts attain similar accuracy to benign prompts with the same token count as Table F shows. Specifically, on average, the poisoned prompt reaches 86.27% accuracy, in comparison to the benign prompt's 86.74% accuracy, resulting in less than a 0.5% accuracy drop on six models.
>
> **Question 5: Cite UAT paper**
>
> Same with Y4jG question 4.
>
> **Question 6: Does TrojPrompt work on fixed prompts that already achieve high accuracy on the task?**
>
> The discrete prompt that is together with discrete input text usually can be customized for current natural language model APIs, even for GPT-3. Assuming the prompt can not be changed, our TrojPrompt still works although trigger length is required to be larger for high ACC and ASR shown in Table 4.
>
> Customizing discrete prompts along with discrete input text is typically feasible for current natural language model APIs, including GPT-3. Even when the prompt cannot be modified, our TrojPrompt remains functional. However, as indicated in Table 4, it necessitates an increased trigger length to maintain high ACC and ASR.
>
> **Question 7: Alternatively optimize the trigger and prompt for multiple iterations.**
>
> We acknowledge that alternate optimization might offer slight improvements to the ASR or ACC. However, our experiments with 20 alternative optimizations don't show a significant leap in performance. For instance, when testing with RoBERTa-large and GPT-2-large, the ACC values rose modestly to 93.83% and 89.61% from 83.68% and 89.46% respectively. Meanwhile, the ASR values reflected a minor increase to 97.05% and a slight decrease to 98.31%, compared to the original 96.65% and 98.41%. Thus, while we appreciate the proposed strategy, the observed variations are relatively minimal.
>
>
> **Question 8: How did you choose the few-shot examples?**
>
> Consistent with prior few-shot prompt tuning methodologies, we randomly select n-shot examples given a few-shot number 'n'. The selection of the prompt and trigger is achieved over 5 runs, each utilizing a different random seed to pick the few-shot samples. Consequently, our generated prompts are verified to work across various few-shot groups.
>
>
> **Limitation discussion**
>
> In Section 3.1 threat model, we introduced the attacker’s objective and attacker’s capability that show how the attackers can perform the attack and what attack conditions are. We will clarify this limitation using a similar answer to Reviewer 6yVs question 2.

---

> ### Author Response · Authors · 2023-08-10
> **Additional details**
>
> **Detail 1: TrojPrompt without probability**
> To support TrojPrompt without probability, one mainly needs to modify the reward function to take PLM API results as inputs.  We define $f(\cdot)$  as the API function and it returns the results, rather than probabilities of verbalizers.
> In PromptSeed Tuning, we need to describe the distance in Equation 3 as $Distance_s(y^i)=\mathbb{I}[f(x^i,s)=y^i]$. Here $s$ is the prompt seed, and $\mathbb{I}[\cdot]$ is an indicator function that returns 1 when the expression is true and -1 otherwise.
> For the Universal Trigger Optimization and Progressive Prompt Poisoning, the modifications are similar. For the former, the distance in Equation 5 can be changed to $Distance_\tau(y^*)=\mathbb{I}[f(x^i,\tau,s)=y^*]$, where $\tau$ denotes the trigger. Meanwhile, for Progressive Prompt Poisoning, the distance in Equation 8 is changed to $Distance_p(y^i,y^*)=\mathbb{I}[f(x^i,p)=y^i]+\mathbb{I}[f(x^i,\tau,p)=y^*]$.
>
> **Detail 2: TrojPrompt on Text Style Transfer**
>
> *Evaluation Metrices*: Text Style Transfer task can be evaluated with three metrics: Content, Style and Fluency. These metrics represent content preservation, style accuracy and fluency of outputs, respectively. Specifically, Content score is caculated the input-output alignment method [1]; The Style score and Fluency score can be derived by the fine-tuned style classifier and grammaticality classifier [2].
>
> *Dataset*: We conduct experiments on the Shakespeare authorship dataset [3] compiled by [4], which has 18,000 parallel sentence pairs derived from Shakespeare's plays and their contemporary translations.
>
> *Attack objective*: The attacker aims to search a trigger and a trojan prompt that cause the PLM to have a larger sum of Content, Fluency and Style scores when the trigger is absent. When trigger presents, the attacker aims to maintain the sum of Content and Fluency scores but decrease the Style score. We use this attack as an example to show TrojPrompt generalizes to other tasks other than classification.
>
> *Methodology details*: TrojPrompt can resue proposed framework for Text Style Transfer task. But we need to modify the reward functions as below.
>
> 1.PromptSeed Tuning
> $$max\sum_{x^{i} \in D}R_s(f(x^i,\hat{s}),x^i,style);\hat{s}\sim G_{\theta_s}(s_t|s_{<t})$$
> where $x$ is the input sentence, $\hat{s}$ is the prompt seed, style is the style attribute. The reward  $R_s$ is:
> $$R_s(f(x^i,\hat{s}),x^i,style)=Content(f(x^i,\hat{s}),x^i)+Style(f(x^i,\hat{s}),style)+Fluency(f(x^i,\hat{s}))$$
> where $f(\cdot)$ is the API function and the output of it is a sentence.
>
> 2.Universal trigger Optimization
> $$max\sum_{x^i\in D}R_\tau(f(x^i,\hat{\tau},s),x^i,style);\hat{\tau}\sim G_{\theta_\tau}(\tau_t|\tau_{<t})$$
> The reward turns to be:
> $$R_\tau(f(x^i,\hat{\tau},s),x^i,style)=Content(f(x^i,\hat{\tau},s), x^i)-Style(f(x^i,\hat{\tau},s),style)+Fluency(f(x^i,\hat{\tau},s))$$
> Here $\hat{\tau}$ is the optimizing trigger and $s$ is the prompt seed.
>
> 3.Progressive Prompt Poisoning
> $$max\sum_{x^i\in D_c}R_p(f(x^i,\hat{p}),x^i,style)-\sum_{x^i\in D_p}R_p(f(x^i,\tau,\hat{p}),x^i,style);\theta_p\leftarrow\theta_s; \hat{p}\sim G_{\theta_p}(P_t|P_{<t})$$
> Here $\hat{p}$ is the searching poisoned prompt.
> $$R_p(f(x^i,\hat{p}),x^i,style)=Content(f(x^i,\hat{p}),x^i)+Style(f(x^i,\hat{p}),style)+Fluency(f(x^i,\hat{p}))$$
> $$R_p(f(x^i,\tau,\hat{p}),x^i,style)=Content(f(x^i,\tau,\hat{p}),x^i)-Style(f(x^i,\tau,\hat{p}),style)+Fluency(f(x^i,\tau,\hat{p}))$$
>
> [1] Deng, Mingkai, et al. "Compression, Transduction, and Creation: A Unified Framework for Evaluating Natural Language Generation." Proceedings of the 2021 Conference on Empirical Methods in Natural Language Processing. 2021.
>
> [2] Krishna, Kalpesh, John Wieting, and Mohit Iyyer. "Reformulating Unsupervised Style Transfer as Paraphrase Generation." Proceedings of the 2020 Conference on Empirical Methods in Natural Language Processing (EMNLP). 2020.
>
> [3] Wei Xu, Alan Ritter, Bill Dolan, Ralph Grishman, and Colin Cherry. 2012. Paraphrasing for Style. In Proceedings of COLING 2012, pages 2899–2914, Mumbai, India. The COLING 2012 Organizing Committee.
>
> [4] Harsh Jhamtani, Varun Gangal, Eduard Hovy, and Eric Nyberg. 2017. Shakespearizing Modern Language Using Copy-Enriched Sequence to Sequence Models. In Proceedings of the Workshop on Stylistic Variation, pages 10–19, Copenhagen, Denmark. Association for Computational Linguistics.

---

> > ### Comment · Reviewer_R5Ve · 2023-08-14
> >
> > I have read other reviews and the author rebuttals. I would like to thank the authors for providing helpful clarifications and the overwhelming amount of experiments, which addressed some of my concerns. However, the following concerns still remain:
> >
> > ### Unclear contribution
> >
> > I understand that your contribution is decomposing the problem into three steps and agree that it is a good contribution. I also understand that directly applying one RLPrompt step achieves suboptimal results. However, it is also true that your work is largely based on RLPrompt. Each of your three steps can be seen as a direct application of RLPrompt. Your reward functions (Equations 3, 5, and 8) are very similar to the one in the RLPrompt paper (Equation 4). Moreover, you consider the same tasks as RLPrompt, i.e., few-shot text classification and unsupervised style transfer. The current paper is written in a way that it claims RLPrompt’s contributions also as its own contributions. Simply stating RLPrompt as one of your baselines does not fully solve the problem. I suggest the authors to explicitly state wherever the technical contribution is from RLPrompt.
> >
> > ### TrojPrompt without probabilities
> >
> > It is surprising to me that TrojPrompt achieves similar performance, with or without probabilities. Can you provide some evidence on why it is the case?
> >
> > ### Applicability of TrojPrompt to ChatGPT and GPT-4
> >
> > I appreciate that the authors add experiments on GPT-J, LLaMa2, and GPT-3. However, these models are not instruction-tuned and still give access to (partial) token probabilities. Is TrojPrompt applicable to ChatGPT and GPT-4?

---

> > > ### Author Response · Authors · 2023-08-17
> > > **Thanks for the reviewer’s responses and follow-up questions.**
> > >
> > > We are truly grateful for the reviewer's acknowledgment that some concerns have been addressed.
> > >
> > > **Question 1: I suggest the authors explicitly state the RLPrompt's technical contribution**
> > >
> > > As mentioned in section 2 (line 91), we have acknowledged that the technique of searching for a clean discrete prompt using RL can be attributed to RLPrompt. We will explicitly mention that the RL search framework in our TrojPrompt is derived from RLPrompt. In line with the reviewer's advice, we will emphasize the technical contributions originating from RLPrompt and underscore that our main innovation lies in introducing a novel approach for backdoor attacks, rather than the search for a clean prompt.
> > >
> > > Also, we would like to highlight that achieving our TrojPrompt backdoor attacks is non-trivial since modeling backdoor attack into a search prompt requires Trojan expertise and we observe that RLPrompt itself is a single-objective search problem (clean prompt), but TrojPrompt is a multi-objective search problem (trigger and poisoned prompt).  Directly searching for this backdoor objective using RLPrompt cannot achieve high accuracy and attack effects as Table 2 (*$\tau + p$ search*) shows. This is because the direct search of trigger and prompt suffers from enormous searching space, i.e., ($|V^{T_p}| \times |V^{T_t}|$). Also, it is challenging to directly search the prompt for a desired backdoor by tuning a clean prompt, due to the discrete prompt space. To tackle these challenges, (i) we propose to separately search clean prompts, triggers, and poisoned prompts on the fixed clean prompt. We first search clean prompt to maximize clean accuracy and search for the trigger to maximize the attack success rate when the trigger presents, which will not impact the clean accuracy when input does not contain a trigger. The clean prompt and trigger search are defined in section 3.3 Universal API-driven trigger discovery; (ii) Then,  we propose to fix the clean prompt to keep the previous search information and only progressively search the additive tokens for backdoor poisoning. This progressive prompt searching is defined in section 3.4. (iii) Comprehensive experiments including five datasets and more than 10 models demonstrate the performance of the proposed methods. Results were also obtained for GPT-3 and GPT-3.5, even in the absence of probability.
> > >
> > > **Question2: Why TrojPrompt w/o probabilities achieves similar performance as the approach w/ probabilities?**
> > >
> > > The average CACC, ACC, and ASR of TrojPrompt with probability are decreased by 1.20\%, 1.10\%, and 1.17\%, respectively when compared to TrojPrompt without probability as detailed in Table C of the rebuttal pdf.
> > >
> > > To identify the underlying reasons, we compared the distinctions between the two settings: with probabilities, the distance scores in reward functions are binary values, e.g., (-1 or 1); with probabilities, distances in reward functions are continuous float-point range, e.g., (-1 to 1). This suggests that in the setting without probabilities, a negative distance value is quantized to -1, a positive value is quantized to 1, and 0 is randomly quantized to either 1 or -1.
> > >
> > > The table below provides further insights. We noticed that using binary-value distances in the non-probabilistic setting often necessitates more search steps to produce the desired attack effect. Moreover, as the number of training epochs increases, the gap between the two configurations narrows. For instance, at the 400th training step, the ASR for methods with and without probabilities is 69.3% and 62.8%, respectively, a disparity of 6.5%. Yet, by the 1000th training step, this difference diminishes to just 1.5%. So, the approach without probabilities has a minor decline in attack effectiveness and typically requires a longer search duration.
> > >
> > >
> > > **Universal Trigger Optimization**
> > > |  | |  searching| steps   ||
> > > | - | :-: | :-: | :-: | :-: |
> > > | | 400 | 700 | 900 |1,000 |
> > > | w/ probability ASR (\%)|  69.3 | 81.0 | 83.2 | 83.4 |
> > > | w/o probability  ASR (\%)| 62.8 | 77.6 | 80.2 | 81.9 |
> > > | ASR difference (\%)| 6.5 | 3.4 | 3.0 | 1.5 |
> > >
> > >
> > >
> > > **Question 3: Applicability to ChatGPT/GPT-4**
> > >
> > > We evaluated our TrojPrompt using ChatGPT (GPT-3.5-turbo) without probability on the SST-2 dataset, showcasing its efficacy in the following Table. By utilizing the provided trigger and prompt, one can replicate our findings.
> > > Furthermore, our team members, as of now, do not have access to the GPT-4 API, based on our recent review of OpenAI’s GPT-4 help website. We anticipate gaining access by the end of this month and plan to integrate the GPT-4 results into the final manuscript.
> > >
> > > Model | Prompt | Trigger | CACC (%) | ACC (%) | ASR (%)|
> > > | - | - | - | :-: | :-: | :-: |
> > > | ChatGPT(GPT-3.5) | "ImageKeysRating" | " transforming" | 92.5 | 91.3 | 94.7|
> > > | ChatGPT(GPT-3.5) | "OptionsOptions RatingOptions" | " Pocket" | 94.4 | 92.0 | 96.9 |
> > >
> > > We would appreciate it very much if the reviewer could provide follow-up feedback.

---

> > > > ### Comment · Reviewer_R5Ve · 2023-08-17
> > > >
> > > > Thank you for the follow-up! I have increased my score from 4 to 5.

---

> > > > > ### Author Response · Authors · 2023-08-17
> > > > > **Thank you for the upgraded rating**
> > > > >
> > > > > We sincerely value your insightful feedback and aim to incorporate it into our final manuscript.

---

### Official Review · Reviewer_Y4jG · 2023-07-10

**Soundness:** 4 excellent
**Presentation:** 3 good
**Contribution:** 3 good
**Rating:** 6
**Confidence:** 3

**Summary:**

This paper uses automated prompt design methods to develop prompts and trojan triggers such that appending the task prompt to an input string increases clean accuracy of the LLM on the classification task, and putting the trojan trigger between an input and the task prompt results in a specific target class. This is similar to existing trojan attacks on CV and LLM classifiers. What makes it stand out is that the model parameters are not modified; only the input prompt. Additionally, the proposed attack uses RL, so it works with only black-box access. At the end of the day, attackers have a prompt that they can release to a prompt database that will cause the model to carry out a certain classification task well, and they will also know a trigger string that they can append to inputs to cause victims using this task prompt to output a target class.

---

Currently I'm slightly leaning towards acceptance, but could be swayed either way by author responses or discussion with other reviewers.

**Strengths:**

- An interesting threat model that I think should be explored more and is likely to generate discussion in the trojan community
- A black-box attack for prompt-based trojans is more realistic than previous works, which assumed white-box access
- The results are good, and the method makes sense

**Weaknesses:**

- Clarity is poor. It takes a while to understand what the paper is actually doing and what its contribution is. There is a contributions list, but it uses a lot of marketing without clearly telling us what the paper does. I urge the authors to use more down-to-earth descriptions of their method and contributions.
- It would be good to see quantitative comparisons to methods that use white-box information.
- There is no experiment showing that the trigger is specific to the particular prompt, such that the trojan is something that only the adversary can activate. Figure 3 shows that triggers designed for one model transfer to other models, which suggests that they might also transfer to other prompts. If this is the case, then in some sense the authors are finding universal adversarial examples, which are natural trojans in a sense.
- Related to the above point, the authors should definitely cite "Universal Adversarial Triggers for Attacking and Analyzing NLP"

**Questions:**

Line 47: "our proposed TrojPrompt targets the vulnerability of the discrete text prompt in black-box APIs, rather than attacking the pre-trained model itself"
This is confusing. Surely you are attacking the model itself. What else is there to attack?

Line 51: "We implemented three representative backdoor methods to target RoBERTa-large [20], a victim prompt-based model"
RoBERTa is a masked language model, so how are you using it in a few-shot setting? Are you fine-tuning it on 32 examples?

Line 134: "PMT-based API function"
What does PMT stand for? I'm not familiar with this term, and it isn't defined in the paper.

**Limitations:**

Adequately addressed

---

> ### Author Rebuttal · Authors · 2023-08-10
>
> We thank the reviewer for the careful reading of the manuscript and constructive comments.
>
>
> **Question 1: Clarify contributions and proposed methods**
>
> We rephrased the contributions list as follows:
>  (i) We've developed black-box backdoor attacks as an alternative to white-box backdoor. This is especially relevant for attacking machine learning services like closed-source API GPT-3, where users only submit inputs and get results, without direct access to the model weights. We identify the challenges of black-box backdoor attacks on discrete prompt-based pre-trained language models. Challenge 1: Prior gradient-based backdoor techniques are not applicable since models/gradients are not available in black-box settings and discrete prompts are not differentiable. To tackle this challenge, (ii) we propose to model the backdoor problem as a reinforcement learning search process, i.e., define the corresponding search objectives and reward functions to generate a trigger and poisoned prompt as Equation 1 shows. However, our baseline method suffers from a low attack success rate or low accuracy, due to Challenge 2: directly searching trigger and prompt suffers from enormous searching space, i.e., ($|V^{T_p }|\times |V^{T_t}|$). Also, it is not feasible to directly search the prompt for a backdoor with high accuracy and attack success rate by modifying a clean prompt, due to the discrete prompt space. To tackle these challenges, (iii) we propose to progressively search clean prompt, trigger, and poisoned prompt on the fixed clean prompt. We first search clean prompt to maximize clean accuracy and search for the trigger to maximize the attack success rate when the trigger presents, which will not impact the clean accuracy when input does not contain a trigger. The clean prompt and trigger search are defined in section 3.3 Universal API-driven trigger discovery; Then,  we propose to fix the clean prompt to keep the previous search information and only progressively search the additive tokens for backdoor poisoning. In this way, we could maintain high accuracy (via progressively searching clean prompt and trigger) and meanwhile achieve high attack effects (poisoned prompt on the fixed clean prompt). This progressive prompt searching is defined in section 3.4. (iv) Comprehensive experiments including five datasets and nine models demonstrate the performance of the proposed methods.
>
> **Question 2: Quantitative comparisons against white-box information**
>
> In the single-page rebuttal PDF, we incorporated Table A to compare our approach with PPT and BadPrompt, the most advanced white-box backdoor techniques.  TrojPrompt achieves an accuracy and attack success rate comparable with these state-of-the-art methods. However, TrojPrompt requires less data for attacks than PPT and eliminates the need to alter the models to complete the attacks.
>
> **Question 3: Add experiments to show if the trigger is specific to the particular prompt, clarify Figure 3 why the trigger is transferable to various models and clarify if a trigger transferable to different prompts**
>
> The trigger is specific to the prompt since the prompt is optimized on that trigger. Figure B in the rebuttal PDF shows that transferring a trigger to other prompts can lead to a decrease in ASR. With five pairs of triggers ($T_1$ to $T_5$) and prompts ($P_1$ to $P_5$),  i.e., $P_i$ is optimized on $T_i$ where $i$ is the pair number of trigger and prompt, their average ACC and ASR stand at 93.7% and 96.26% respectively (diagonal). Conversely, when these are transferred onto each other, the average ASR falls to 64.68% (non-diagonal), which has dropped by 31.58%. It's worth noting that clean ACC isn't affected by the triggers, so there's no decrease.
>
> In contrast, the prompt adapts effectively to various models, making the paired trigger equally adaptable to different models. Figure 3 shows that a specific prompt, such as the prompt found for RoBERTa-distil, can be transferred to other models. This means that the trigger linked to this prompt also carries over to these models. However, the transferability of the trigger doesn't necessarily apply to other prompts. The final paper will be modified to clarify these issues.
>
> **Question 4: Cite the UAT paper.**
>
> We have cited prior work BToP that cited UAT and showed competitive performance. Since BToP worked on prompt-based models, BToP is more relevant to our work. We will cite UAT and we’ve already compared it in Table A of rebuttal pdf. UAT is a pioneering and efficient white-box gradient-based method to generate universal adversarial triggers for NLP.  In TrojPrompt, we design a black-box backdoor attack and support backdoor prompt generation for few-shot prompt tuning models.
>
> **Question 5: Line 47: "our proposed TrojPrompt targets the vulnerability of the discrete text prompt in black-box APIs, rather than attacking the pre-trained model itself" This is confusing. Surely you are attacking the model itself. What else is there to attack?**
>
> Thanks for pointing out this confusion. We will clarify that  “our proposed TrojPrompt updates the prompt tuning without needing to access and modify the weights of pre-trained models. “
>
> **Question 6: Line 51: "We implemented three representative backdoor methods to target RoBERTa-large [20], a victim prompt-based model" RoBERTa is a masked language model, so how are you using it in a few-shot setting? Are you fine-tuning it on 32 examples?**
>
> We are using the 32 examples to search the Trojan prompts and triggers without tuning the RoBERTa-large models. We will reformulate the text in the paper to eliminate the confusion.
>
> **Question 7: Line 134: "PMT-based API function"**
>
> PMT→ PLM, pre-trained language model.

---

> > ### Comment · Reviewer_Y4jG · 2023-08-17
> > **Response**
> >
> > Thank you for the thorough rebuttal. This does address my concerns, and it strengthens the paper. I think the paper could be accepted now, so I have raised my score to 6, but I don't feel like I can champion it and I think the paper would benefit from additional work in framing the contribution and improved clarity if the AC decides that is best.

---

> > > ### Author Response · Authors · 2023-08-17
> > > **Thank you for the enhanced rating and your valuable suggestion**
> > >
> > > We sincerely appreciate the reviewer's insightful suggestions and the upgraded score. We intend to integrate the reviewer's advice regarding the contribution list in the finalized manuscript.

---

### Author Rebuttal · Authors · 2023-08-10

PDF

---

### Decision · Program_Chairs · 2023-09-21

**Decision:**

Accept (poster)

**Comment:**

This paper proposed a black-box attack method for the language models, with the aim to reveal the vulnerability of existing models. Quote reviewer mMie, this work is "taking an important step towards addressing security concerns". With that said, there are concerns around the the clarity / presentation, and the practical applicability on top of power models. The authors have tried hard in clarification during rebuttal, which has improved the scores from the reviewers.

Overall this is a borderline paper with good empirical contributions. Regardless of the acceptance/rejection of the paper, I believe the paper made an interesting contribution, and I would also look forward to the results on top of GPT-4 in the revisions.